# How Do Neighbourhood Definitions Influence the Associations between Built Environment and Physical Activity?

**DOI:** 10.3390/ijerph16091501

**Published:** 2019-04-28

**Authors:** Suzanne Mavoa, Nasser Bagheri, Mohammad Javad Koohsari, Andrew T. Kaczynski, Karen E. Lamb, Koichiro Oka, David O’Sullivan, Karen Witten

**Affiliations:** 1SHORE and Whariki Research Centre, School of Public Health, Massey University, P.O. Box 6137, Auckland 1141, New Zealand; K.Witten@massey.ac.nz; 2Melbourne School of Population and Global Health, The University of Melbourne, Melbourne, VIC 3010, Australia; 3The Visualisation and Decision Analytics (VIDEA) lab, Centre for Mental Health Research, Research School of Population Health, College of Health and Medicine, The Australian National University, Canberra, ACT 2601, Australia; nasser.bagheri@anu.edu.au; 4Faculty of Sport Sciences, Waseda University, Saitama 359-1192, Japan; Javad.Koohsari@baker.edu.au (M.J.K.); koka@waseda.jp (K.O.); 5Behavioural Epidemiology Laboratory, Baker IDI Heart and Diabetes Institute, Melbourne, VIC 3004, Australia; 6Prevention Research Center, Arnold School of Public Health, University of South Carolina, Columbia, SC 29208, USA; ATKACZYN@mailbox.sc.edu; 7Murdoch Children’s Research Institute, Melbourne, VIC 3052, Australia; karen.lamb@mcri.edu.au; 8School of Geography, Environment and Earth Sciences, Victoria University, Wellington 6012, New Zealand; david.osullivan@vuw.ac.nz

**Keywords:** neighbourhood, scale, built environment, physical activity, walking

## Abstract

Researchers investigating relationships between the neighbourhood environment and health first need to decide on the spatial extent of the neighbourhood they are interested in. This decision is an important and ongoing methodological challenge since different methods of defining and delineating neighbourhood boundaries can produce different results. This paper explores this issue in the context of a New Zealand-based study of the relationship between the built environment and multiple measures of physical activity. Geographic information systems were used to measure three built environment attributes—dwelling density, street connectivity, and neighbourhood destination accessibility—using seven different neighbourhood definitions (three administrative unit boundaries, and 500, 800, 1000- and 1500-m road network buffers). The associations between the three built environment measures and five measures of physical activity (mean accelerometer counts per hour, percentage time in moderate–vigorous physical activity, self-reported walking for transport, self-reported walking for recreation and self-reported walking for all purposes) were modelled for each neighbourhood definition. The combination of the choice of neighbourhood definition, built environment measure, and physical activity measure determined whether evidence of an association was detected or not. Results demonstrated that, while there was no single ideal neighbourhood definition, the built environment was most consistently associated with a range of physical activity measures when the 800-m and 1000-m road network buffers were used. For the street connectivity and destination accessibility measures, associations with physical activity were less likely to be detected at smaller scales (less than 800 m). In line with some previous research, this study demonstrated that the choice of neighbourhood definition can influence whether or not an association between the built environment and adults’ physical activity is detected or not. This study additionally highlighted the importance of the choice of built environment attribute and physical activity measures. While we identified the 800-m and 1000-m road network buffers as the neighbourhood definitions most consistently associated with a range of physical activity measures, it is important that researchers carefully consider the most appropriate type of neighbourhood definition and scale for the particular aim and participants, especially at smaller scales.

## 1. Introduction

Many studies have investigated associations between the neighbourhood-built environment attributes and the physical activity of residents, with evidence accumulating on the health benefits of living in higher density neighbourhoods with well-connected street networks and pedestrian access to a range of amenities [1,2,3]. Within this area of research, an important methodological challenge is how to define a “neighbourhood” [4,5,6]. A neighbourhood refers to the geographical area within which environmental attributes are investigated in relation to physical activity. It is hypothesised that residents are only able to walk within such geographical areas of their homes. The challenge of defining a neighbourhood is also shared by the wider neighbourhood research field and has been regularly highlighted over the past decade [4,5,7,8,9]. Currently, researchers have been using a variety of neighbourhood definitions and there is little consensus as to the most appropriate geographic scales [10,11]. This is a problem because using different neighbourhood definitions can change the results [5,10,11,12,13,14,15] and the lack of standardisation makes it difficult to compare and combine evidence across studies [10]. Furthermore, identifying the most appropriate geographical scales at which built environments may influence health behaviours is an important step in translating the evidence into urban design and public health practice [16]. For example, evidence that the proximity of public open spaces is associated with better health behaviours and outcomes is useful but not sufficient for those that (re)design the built environment or write environmental policy. Urban design policy makers and practitioners also need to know how far away the public open spaces need to be located from people’s homes to maximise their amenity value and health outcomes. In other words, a better understanding of the distances and geographical scales at which the built environment influences health could inform more effective urban design and policy interventions [10,16,17].

Neighbourhood is commonly conceptualised as the home neighbourhood and operationalised using geographic information systems (GIS). The dimensions of a neighbourhood are determined by the type of boundary applied and its size or geographical scale. There are three main types of neighbourhood definitions used in built environments and public health research: administrative units, circular buffers, and road network buffers. Administrative units (e.g., census tracts, postal codes) allow researchers to link their data with secondary data sources. However, they are subject to the modifiable areal unit problem (MAUP) where results can vary depending on the division of the study area, a zonation or aggregation effect [18], and the size of the units used, a scale effect [19]. Circular (or Euclidean) buffers are circles of a defined radius centred on an address, whereas road network buffers are calculated by drawing an area around an address that is accessible by travelling a defined distance along roads. Circular buffers are simpler to calculate, but road network buffers are conceptually more appealing because they better represent where people may travel, particularly in areas with features such as rivers, lakes, or a poorly connected road network [20]. Both circular and road network buffers address the zonation effect of the MAUP but are still subject to scale effects. A further issue is that appropriate geographical scales are likely to vary for different population groups, different built environment measures, and different outcomes [4,5]. In practice, built environment and physical activity researchers use a variety of geographical scales ranging from 100–8050 m [4].

The impact of neighbourhood definition on research results has been frequently demonstrated in the wider literature, and more recently in several studies focused on the built environment and physical activity [10,12,15,21]. While researchers have struggled to identify optimal neighbourhood definitions, some of the existing built environment and physical activity research suggests that larger buffers might best explain objectively measured moderate-to-vigorous physical activity in children [21], and self-reported walking for transport in adults [10]. Existing research has tended to focus on single measures of physical activity, yet it is possible that the choice of neighbourhood definition may differ for different aspects of physical activity. Furthermore, appropriate neighbourhood definitions are likely to vary for different aspects of the built environment and in different contexts. Therefore, it is important to investigate this issue across a range of exposure measures, outcome measures (e.g., walking for transport, walking for recreation moderate–vigorous physical activity, overall physical activity), population groups, and locations.

Given these considerations, the main purpose of this paper is to test the hypothesis that neighbourhood definitions of different types and geographical scales will determine whether or not evidence of an association between the built environment and physical activity is identified in statistical models. While this question has been explored in existing studies, to our knowledge no study has examined this issue in the context of associations between the built environment and objectively measured physical activity in adults, nor has any research yet examined how the choice of neighbourhood definition differentially impacts associations between different physical activity measures (e.g., self-report versus objective, recreational versus transport walking). Therefore, we also test the hypothesis that detection of an association will vary with different neighbourhood definitions, built environment measures, and physical activity measures.

## 2. Materials and Methods 

### 2.1. Study Background

This study is based on data collected within the Understanding the Relationship Between Physical Activity and Neighbourhood (URBAN) study [22]. The URBAN is part of the International Physical Activity and Environment Network (IPEN) Adult study, an observational, cross-sectional study in 12 countries [23]. The present study uses data from New Zealand and uses IPEN protocols for exposure and outcome measures [24,25]. Ethical approval was granted by the Auckland University of Technology and Massey University ethics committees (AUTEC: 07/126, MUHECN: 07/045). All participants provided written informed consent. 

The URBAN study recruited 2033 adults aged 20–65 years from 24 high- and 24 low-walkability neighbourhoods in four New Zealand cities: Waitakere, North Shore, Wellington and Christchurch (12 neighbourhoods in each city). Compared to the other eleven IPEN countries, the neighbourhoods in these four New Zealand cities tended to be less walkable with lower street connectivity (i.e., more cul-de-sacs), lower residential density, and lower land-use mix [25]. Walkability scores were calculated for each census meshblock using GIS and included measures of residential dwelling density, street connectivity, land-use mix, and retail floor area ratio [22]. The meshblock is the smallest spatial statistical unit in New Zealand, containing on average 110 people in urban areas and varying in size. Meshblocks with walkability scores in the highest tertile were defined as highly walkable, while meshblocks with walkability scores in the lowest tertile were defined as having low walkability. Meshblocks with average walkability scores were deliberately excluded to maximise variability as prescribed by IPEN protocols. The URBAN study neighbourhoods comprised five-plus contiguous meshblocks with consistently high- or low-walkability scores. All eligible neighbourhoods within each city were identified. Where there were more neighbourhoods than required, the URBAN study research team purposefully selected neighbourhoods based on local knowledge. 

The 48 neighbourhoods were mainly suburban (*n* = 36) and dominated by residential land use with large areas of open space. Ten neighbourhoods were located within or adjacent to an activity centre with a mix of land uses—including retail, open space, institutional, and light industrial—but still dominated by residential land use. Two neighbourhoods were located on the outskirts of the Wellington central business district (CBD). These neighbourhoods were dominated by a mix of retail, commercial, institutional, and light industrial, with residual areas of residential and open space land use. Table 1 presents summary statistics describing the neighbourhood demographics. The NZ deprivation score is a value between 1 and 10, with 10 indicating that an area is more deprived [26]. The CBD neighbourhoods were more deprived and had a larger number of people and dwellings than suburban and activity centre-based neighbourhoods.

For this study, 44 participants were excluded due to accelerometer exclusion criteria (see below) and the inability to locate residential addresses, leaving a total of 1989 participants. The methodology relevant to the present paper is described below. Detailed methods and participant demographics for the broader study are described elsewhere [22,27].

### 2.2. Physical Activity Measures

Objective physical activity was measured using Actical accelerometers (Mini-Mitter, Sunriver, OR, USA), which participants wore on their hips for seven consecutive days during waking hours. Accelerometers sense frequency and intensity of movement [28] and can distinguish between less intense physical activity such as walking and more intense physical activity such as riding a bicycle or running. 

The accelerometers were set to record every 30 seconds. The raw output from the accelerometer is a unitless measure called a count [29], with higher counts indicating more intense physical activity. Periods of greater than 59 minutes of consecutive zero counts (indicating likely non-wear time) or where the accelerometer was worn for less than 60 minutes were excluded from analysis. Days with less than 10-hours-per-day wear time were also excluded. 

Self-report physical activity data were collected using the International Physical Activity Questionnaire—Long Form (IPAQ-LF; [30]). Three self-reported measures of physical activity measures were created based on this questionnaire: self-reported walking for transport, self-reported walking for recreation, and total self-reported minutes walking for all purposes.

### 2.3. Neighbourhood Definitions

Overall, seven different neighbourhood definitions were created for each participant at a range of geographical scales. Three of the seven areas were based on the administrative units: the meshblock, the census area unit, which is comprised of meshblocks in urban areas and contains between 3000—5000 people [31], and the URBAN study neighbourhoods (see Table 1 for relative neighbourhood sizes). The four remaining neighbourhood definitions were road network buffers centred on participants’ geocoded residential addresses and calculated at four geographical scales commonly used in built environment and health research [4]: 500 m, 800 m, 1000 m and 1500 m. The road network buffers were created using the Service Area function in ArcGIS version 9.3 (ESRI, Redmond, WA, USA) [32]. The road network was supplied by territorial authorities and excluded pedestrian-only paths due to a lack of data. Roads that are inaccessible to pedestrians (i.e., motorways and motorway on and off ramps) were removed prior to analysis. The relative sizes of the neighbourhoods are illustrated in Figure 1, which shows the different neighbourhood definitions for an exampleparticipant.

### 2.4. Built Environment Attributes

Three built environment attributes—dwelling density, street connectivity, and destination accessibility—were calculated for each participant within each of the seven neighbourhood definitions. These three attributes were chosen because they have been frequently found to be associated with physical activity within many studies across different contexts [33,34,35], and they were also shown to be associated with physical activity within the same dataset [27].

Dwelling density was calculated by dividing the number of occupied private dwellings by the residential land area, which was obtained from zoning datasets provided by territorial authorities. Dwelling numbers were sourced from the 2006 New Zealand census at the meshblock level. Since meshblock boundaries align with all administrative neighbourhoods, the number of private occupied dwellings is easily calculated for this type of neighbourhood definition. However, meshblock boundaries do not align with road network buffer boundaries. Therefore, the number of private occupied dwellings within each road network buffer was estimated by calculating a weighted average based on the land area of contributing meshblocks. Street connectivity was calculated by dividing the number of intersections (three or more ways) within the neighbourhood by the area in square kilometres. The calculation of dwelling density and street connectivity measures followed IPEN GIS protocols [24].

Destination accessibility was assessed using the neighbourhood destination accessibility index (NDAI; [27,36]). The NDAI is a measure of access to 31 neighbourhood destinations in eight domains (education, transport, recreation, social and cultural, food retail, financial, health, and other retail). The destination data used to calculate the NDAI were obtained from a range of sources including government (New Zealand Ministry of Education, New Zealand Ministry of the Environment and Land, New Zealand Ministry of Health, Territorial authorities, Liquor Licensing Authority), private spatial data suppliers (Terra Link International, GeoSmart) and an online business directory (www.zenbu.co.nz).

Most NDAI domains were calculated by assigning a score based on the presence of destinations within a neighbourhood. However, the transport and recreation domain scores were based on the density of destinations. The final NDAI score was calculated by summing the weighted domain scores, producing a value between 0 and 31, with a higher score representing better walking access to services and amenities. Since the NDAI was based on the presence/absence of destinations, it will necessarily increase with increased neighbourhood size.

### 2.5. Demographics, Neighbourhood Preference and Neighbourhood Socioeconomic Deprivation

Information on participants’ age, gender, ethnicity, marital status, household income, educational qualifications, occupation, household car access, and preferences for living in a more or less walkable neighbourhood were collected in face-to-face computer-assisted personal interviews (CAPI). 

Individuals may choose to live in neighbourhoods that support physical activity, introducing the possibility that individual neighbourhood preference may confound associations between neighbourhood environments and physical activity [27]. Therefore, neighbourhood preference was measured using items developed by Levine et al. [37]. Participants were provided with illustrations and verbal descriptions of two types of neighbourhoods—a lower-density suburban neighbourhood with common destinations accessible by car and a higher-density urban neighbourhood with most destinations accessible by walking or public transport. Participants indicated which of the two neighbourhood types they would prefer to live in—assuming similar housing costs, school quality and a mix of people in both neighbourhoods—using a five-point scale (strongly prefer walkable, moderately prefer walkable, neutral, moderately prefer less walkable, strongly prefer less walkable) [37].

Neighbourhood socio-economic deprivation was measured using the New Zealand Deprivation Index 2006 provided at the meshblock level [26].

### 2.6. Statistical Analysis

The associations between the built environment and physical activity measures were modelled using linear multi-level mixed-effect models to take into account the clustering of individuals within neighbourhoods (defined as the URBAN study neighbourhood) and cities. The multi-level mixed-effect model was chosen to assess the effect of neighbourhood characteristics on individual physical activity level. The appropriateness of the multilevel structure was tested by applying the likelihood ratio (LR) test to compare an empty model with and without adjustment for clustering (URBAN study neighbourhood nested in cities). The model’s fit was significantly improved (*p* < 0.001) with the inclusion of the neighbourhood level variables, so the multilevel structure was maintained. The association between built environment exposures and physical activity were assessed through three models and progressively adjusted for confounders. All outcome variables were log transformed to have a normal distribution and aid comparison across models. The regression coefficients when exponentiated are the ratio or relative change in the outcome measure for each unit change in the exposure variable. Therefore, regression coefficients were exponentiated and reported as a relative change in the results.

The association between each of the three built environment measures and each of the five physical activity measures was modelled separately for each of the seven neighbourhood definitions. Each association was assessed by adjusting for individual-level factors (sex, age, ethnicity, income, marital status, education, employment and car access), neighbourhood socioeconomic deprivation and neighbourhood preference. All models reported in this paper were fully adjusted for these covariates. Adjusted intraclass correlation coefficients (ICCs) were calculated for null models (constant term in the fixed part) for each outcome. The goodness-of-fit of each model was estimated by calculating the marginal R^2^ (proportion of variance explained by fixed factors alone) and conditional R^2^ (proportion of variance explained by both fixed and random effects) [38]. Statistical analyses were conducted in R [39] using the “lme4” package to fit the linear mixed models and the “MuMIn” package to calculate goodness-of-fit [40,41].

## 3. Results

Descriptive statistics for the outcome measures are presented in Table 2. To put the mean accelerometer counts per hour measure into context: a participant who is washing dishes for an hour might record counts in the order of 600 (~10 counts per minute), while a participant who is continuously playing basketball for an hour might record counts in the order of 282,000 (~4700 per minute) [42]. 

Descriptive statistics for the size of the seven neighbourhood definitions are shown in Table 3. The meshblock is the smallest neighbourhood, with a median area almost one quarter the size of that of the next smallest area (500-m road network buffer). The URBAN study neighbourhood is closest in size to the 500-m road network buffer, and the census area unit falls between the 1000-m and 1500-m road network buffers.

Table 4 displays the descriptive statistics for the built environment measures for each neighbourhood definition. The median street connectivity and dwelling density measures decreased consistently with increasing neighbourhood size. In contrast, NDAI measures consistently increased with increasing neighbourhood size. This is expected because the NDAI measure is calculated solely on the presence and number of destinations, meaning that an increase in neighbourhood size will always result in either no change or an increase in the NDAI score.

The results of the fully adjusted models presented in Table 5 indicate that whether or not an association between the built environment and physical activity was detected depends on the choice of neighbourhood definition, built environment measure and physical activity measure. Coefficients for all models are provided in Appendix A. For all models, the URBAN study neighbourhood and city were modelled as random effects and all other explanatory variables were modelled as fixed effects. The results are reported as the percentage change in the physical activity measure unit per unit increase in the built environment measure (Table 5). Bold text indicates results where there was some evidence (confidence intervals did not cross zero) to support an association between the built environment and physical activity. As expected in built environment research, the effect sizes were small as individual outcomes are more strongly associated with individual predictors.

There was no single neighbourhood definition that resulted in statistical evidence of an association between all built environments and physical activity measures. The meshblock, 500-m and 800-m road network buffers consistently resulted in evidence of an association between dwelling density and all five physical activity measures. For street connectivity, the URBAN neighbourhood, census area unit, and 1000-m road network buffer produced consistent evidence of an association with physical activity. In contrast, there was no single neighbourhood definition that resulted in consistent evidence of an association between NDAI and all five physical activity measures. The neighbourhood definitions where NDAI was most consistently associated with physical activity were the census area unit, 800-m road network buffer and 1000-m road network buffer. Overall, associations between the built environment and physical activity measures were most consistently detected when the 800-m and 1000-m road network buffers/were used.

When comparing models with the same built environment and physical activity measure, the marginal and conditional R^2^ values were similar. This indicates that the choice of neighbourhood delineation did not meaningfully change the amount of variance explained by the models.

## 4. Discussion

The main aim of this paper was to test the hypothesis that neighbourhood definitions of different types and geographical scales will determine whether or not evidence of an association between the built environment and physical activity is captured in statistical models. Extending existing research, this paper also makes a new contribution by examining whether or not the choice of the physical activity outcome measure also determines whether or not an association is detected. Looking first at the individual models, in general, the magnitude of the effects appears meaningful. For a one dwelling per hectare (dph) increase in dwelling density, the estimates ranged from a 0.63% to 1.18% increase in overall physical activity. As a whole, the size of these effects are meaningful when you consider that the median dwelling density of the neighbourhoods in this study (~10 dph) falls within the “low-density suburban” category (8–12 dph) and to reach the next highest density category would require an increase in the order of 5 dph [43]. An increase of this magnitude would be associated with an increase in overall physical activity in the order of 5%. Although the effect sizes for street connectivity were smaller (0.27% to 0.48%), they also represent a meaningful increase in physical activity given that this is associated with increasing street connectivity by one intersection per square kilometre. For NDAI, the effect sizes (0.60% to 0.88%) relate to a one unit increase in NDAI score, which means adding one more different type of destination within the neighbourhood. For example, adding a convenience store to a neighbourhood where there are currently no convenience stores.

Results from this study supported the main hypothesis: that the choice of neighbourhood definition can determine whether evidence can be found or not. For all three built environment attributes there was evidence of an association for at least one of the seven neighbourhood definitions, yet for both street connectivity and NDAI some neighbourhood definitions had no evidence of an association between the built environment and physical activity. Our results also demonstrated that the choice of built environment and physical activity measures also determined whether or not evidence of an association was found. A neighbourhood delineation that is appropriate for one built environment measure may not be appropriate for all built environment measures. Similarly, different delineations may be more appropriate for different physical activity outcome measures. Therefore, it is important to carefully choose neighbourhood definitions and to report results at a range of geographical scales [4]. Similar to previous research, we were unable to clearly identify a single optimal neighbourhood definition for use in the built environment and physical activity research. However, our study showed that associations between the built environment and physical activity were most consistently detected when the 800-m and 1000-m road network buffers were used.

The lack of evidence of association at the smaller geographical scales makes sense when we consider the different types of measure. Given the neighbourhoods included in the study, we would expect dwellings to be the most common feature and for dwellings to be present at all scales. Therefore, it is not surprising that associations between dwelling density and physical activity were found at the smallest scales. In contrast, the NDAI measure is based on the presence of destinations, which are far less common than dwellings, especially in the study neighbourhoods which were largely suburban.

There may be other explanations for the lack of identified associations at the smallest geographical scales. There is a greater effect of positional accuracy (geocoding and spatial data precision and error) at smaller scales [44]. Furthermore, it is possible that smaller scale neighbourhoods are more relevant to population groups not considered in this study (e.g., non-drivers compared to drivers or children compared to adults). For example, in a study of geographic area and scale on the relationship between food environment and behaviour, Thornton and colleagues [11] found no evidence of an association between the food environment at the smallest geographic area (400-m road network buffer) for the full sample, although it reached significance when only households without cars were assessed; a finding that is consistent with travel survey data that shows that people in non-car households are more likely to use active transport modes than households with access to a car [45,46,47].

In general, our findings are consistent with those of similar studies investigating the impact of different neighbourhood boundaries on the built environment and physical activity. Clark and Scott [44] who, in a study of the MAUP on the relationship between the built environment and active travel, concluded that while the choice of neighbourhood definition influences coefficient magnitudes and significance, the patterns were inconsistent for different built environment measures. Our findings that the smaller scales were less likely to detect evidence of an association are similar to studies that suggest that larger buffer sizes might be more appropriate when investigating adults walking for transport [10] and children’s moderate-to-vigorous physical activity [21].

As mentioned earlier, it has been recommended that researchers report GIS-based built environment measures at a range of scales [4], and our results support this. Not only would this assist with greater consistency and comparison across studies, but it would also help identify optimal built environment thresholds to support health behaviour for a range of built environment measures, population groups and health behaviours and outcomes [16]. However, reporting results at a range of scales may be difficult from a practical perspective. Calculating GIS-based measures of the built environment requires technical staff, specialist software, and sufficient computing power. This can make the calculation of built environment measures at a range of geographical scales prohibitively difficult and expensive. Possible solutions to this problem include sharing GIS resources and knowledge (e.g., sharing scripts and GIS-based models that automatically calculate built environment measures, developing manuals) [48], and the provision of open source tools to calculate built environment measures [49]. 

Reporting results at a range of geographical scales does not preclude first determining what scales and ranges are appropriate. An important first step is to consider available theoretical and conceptual models that could assist with decisions about what scales are likely to be most relevant [50]. Other data—such as time-use data [51], travel survey data [52,53], GPS data [54,55], public participation GIS [56] and studies on perceived neighbourhood sizes [57,58]—can also be used to inform the choice of scale by providing information on distances people travel and places they spend time. 

### Limitations

One of the strengths of this study was the use of an objective measure of physical activity, thus avoiding some of the issues with self-report measures such as poor respondent memory, recall bias and under-estimation of incidental activities [59]. However, this study had several limitations. First, a limitation of this outcome measure is that the built environment was assessed for the residential neighbourhood, yet the physical activity data were collected everywhere participants went, not only in their residential neighbourhood (non-context specific). Focusing only on the residential neighbourhood is a common issue in the built environment and health research as neighbourhoods are typically defined around the home address. A related issue is that the geographic context in which the built environment influences physical activity behaviours is unknown. This is the uncertain geographic context problem [60], and it means that neighbourhood definitions such as administrative units and road network buffers may not align with the true context whereby the built environment influences physical activity. In response, there have been calls to include the built environments of non-residential neighbourhoods, such as work and school neighbourhoods [61,62], to move from place-based research to person-based research [11,63], and to move to individualised measures of the built environment [11,60].

A limitation that is more important to the conclusions of the study is that the maximum scale did not go beyond 1500 m. It is possible that there is an even larger scale at which the effect of the built environment on physical activity changes. In other words, while our results suggest that buffers ranging from 800–1500 m are likely to be appropriate, it is possible that scales beyond 1500 m are also appropriate. However, as the scale increases, the reduced heterogeneity may lead to difficulty detecting effects [4,11].

There were methodological limitations related to the incomplete representation of where people can travel and imprecise representation of destinations. When creating road network buffers, a lack of pedestrian network data meant that our study used road network data to represent where people can travel. However, this is an incomplete representation of potential travel paths because it excludes non-road networks that people commonly travel along (e.g., pedestrian-only paths, cycle trails). Therefore, the neighbourhood definitions based on road network buffers are likely only subsets of the experienced neighbourhoods. Research has demonstrated that including pedestrian networks can increase the size of the neighbourhood [64,65] and so we would expect that our road network buffers are underestimations of the size of the neighbourhood accessible within a certain distance. Although the importance of including pedestrian paths when defining neighbourhoods has been identified [64,65], the lack of pedestrian network data makes this challenging in practice. While we were not able to include non-road networks in our study, it is likely that pedestrian network data will become increasingly available with the continual development of freely available OpenStreetMap (OSM) data, and also the development of new methods to approximate footpath locations [66] or extract footpaths from remotely sensed imagery [67,68].

Finally, our study was limited by the imprecise representation of destination data. The location of each destination was represented by a single point, whereas in reality, destinations cover areas of varying sizes and in the case of a large park, several access points are likely. This means that compared to administrative units, road network buffers are less likely to accurately capture destinations when they are represented as points. 

Future research could address some of these limitations by consider individual factors (e.g., non-car households, bicycle ownership) that might be important in determining the scale at which the built environment influences health outcomes and behaviours. Other methods of measuring the built environment should also be considered. For example, kernel density measures are an underutilized technique in built environment and health research that account for the proximity of built environment features to one another [69]. Yet, in a recent food environment study they showed stronger associations with food behaviours than measures calculated using circular or road network buffers [11]. 

## 5. Conclusions

In summary, this study demonstrated that the choice of neighbourhood definition can influence whether or not an association between built environment attributes and adults’ physical activity is detected. Furthermore, the association with physical activity was robust enough to be detected at a range of scales for all built environment measures. Although like previous researchers, this study was unable to identify a single optimal neighbourhood definition, we did note that associations were less likely to be found when measured using smaller neighbourhoods. The 800-m and 1000-m road network buffers were the neighbourhood definitions where associations between the built environment and physical activity were most consistently detected.

It is important that researchers carefully consider the most appropriate type of neighbourhood boundary and geographical scale. To assist in this decision, more evidence on appropriate neighbourhood types and scales is needed not only in different environments, but also with different population groups, built environment measures, outcome measures, scales and neighbourhood definitions. Given the difficulties in trying to identify a single optimal neighbourhood definition and the policy need for evidence to be provided with an associated scale, we suggest that future work of this nature might aim to identify a range of appropriate neighbourhood definitions. Furthermore, future work should compare a greater range of scales than studied here, especially larger scales.

## Figures and Tables

**Figure 1 ijerph-16-01501-f001:**
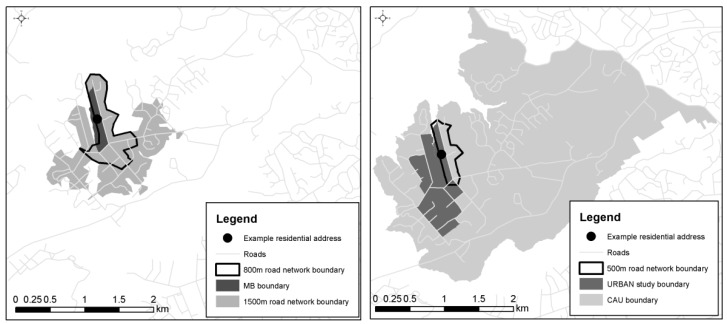
An example of neighbourhood boundaries for a participant. Road data shown in this figure were sourced from Land Information New Zealand (Creative Commons Attribution 3.0 New Zealand) and the neighbourhood boundary data were created as part of this study.

**Table 1 ijerph-16-01501-t001:** Summary of neighbourhood demographics.

Neighbourhood Type	Average Usual Resident Population 2006 (Range)	Average Number of Occupied Dwellings 2006 (Range)	Average NZ Deprivation Score (Range)
Suburban neighbourhoods (*n* = 36)	773.4 (435–1218)	282.1 (153–414)	3.8 (1–10)
Neighbourhoods near activity centres (*n* = 10)	811.5 (495–1251)	306.9 (192–561)	4.8 (1–10)
central business district (CBD) neighbourhoods (*n* = 2)	898.5 (771–1026)	405 (357–453)	7.9 (6–10)

**Table 2 ijerph-16-01501-t002:** Descriptive statistics for the physical activity outcome measures assessed over a 7-day period.

Physical Activity Outcome	Low Walkability			High Walkability			Adjusted Intraclass Correlation Coefficient (ICC) for Null Model
	Mean	Median	SD	Mean	Median	SD	
Self-reported walking for transport (total minutes)	80.0	40	125.2	109.4	50	154.7	0.136
Self-reported walking for recreation (total minutes)	82.5	30	125.0	83.8	30	128.6	0.011
Self-reported overall walking (total minutes)	161.8	100	191.6	192.4	120	220.5	0.120
Mean accelerometer counts per hour	8701.1	8040.0	4215.0	9426.6	8586.7	4692.6	0.048
% time spend in moderate-vigorous physical activity (MVPA)	12.3	11	6.6	12.5	11	6.9	0.080

**Table 3 ijerph-16-01501-t003:** Neighbourhood boundary size descriptive statistics.

Boundary TYPE	Neighbourhood Boundary	*N*	Median (km^2^)	Range (km^2^)	IQR ^a^ (km^2^)
Administrative unit	Meshblock	272	0.05	1.43	0.05
Contiguous administrative units	URBAN neighbourhood	48	0.30	1.03	0.20
Administrative unit	Census area unit	67	1.83	8.96	1.37
Road network buffer	500-m road network buffer	1989	0.28	1.03	0.13
Road network buffer	800-m road network buffer	1989	0.64	0.98	0.31
Road network buffer	1000-m road network buffer	1989	1.00	1.63	0.51
Road network buffer	1500-m road network buffer	1989	2.26	3.41	0.95

^a^ Interquartile range (IQR), ^b^ Understanding the Relationship between physical Activity and Neighbourhood (URBAN) study.

**Table 4 ijerph-16-01501-t004:** Built environment descriptive statistics for neighbourhood boundaries (*n* = 1989 adults).

Neighbourhood Boundary	Dwelling Density (dwellings/Ha)			Street Connectivity (Intersections/km^2^)			Neighbourhood Destination Accessibility Index (NDAI) (Score 0–31)		
	Median	Range	IQR	Median	Range	IQR	Median	Range	IQR
Meshblock	11.9	0.6–80.8	8.8–15.4	25.4	0–311.4	3.2–48.0	2.0	0–10.8	0.6–5.0
URBAN neighbourhood	11.8	2.1–58.3	8.0–15.1	33.2	3.7–111.7	14.9–40.2	5.9	2.5–18.9	4.2–8.1
Census area unit	8.8	1.3–32.1	5.8–11.0	25.6	3.6–92.3	15.3–33.8	9.3	2.2–24.1	6.1–13.4
500-m road network buffer	10.2	1.1–42.0	8.4–12.5	34.1	0–101.1	24.8–42.5	6.4	0–24.6	4.1–9.4
800-m road network buffer	9.8	1.9–37.3	8.4–11.8	32.5	0–91.2	25.6–39.8	10.2	0–29.5	6.2–14.9
1000-m road network buffer	9.6	2.4–36.6	8.4–11.4	31.1	0–90.6	25.6–38.7	13.9	0–34.5	7.9–19.4
1500-m road network buffer	9.3	2.5–33.0	8.3–11.0	29.6	0–76.5	25.3–38.2	20.7	0–40.2	14.6–7.4

**Table 5 ijerph-16-01501-t005:** Percentage change (95% CI) in fully adjusted models of physical activity, for a one unit change in the built environment measures for the seven neighbourhood boundaries. All models were fully adjusted for sex, age, ethnicity, income, marital status, education, employment, car access, neighbourhood socioeconomic deprivation and neighbourhood preference. Bold text indicates results where there was some evidence (confidence intervals did not cross zero) to support an association between the built environment and physical activity.

Built Environment Measure	Neighbourhood Definition	Mean Accelerometer Counts/Hour % Change (95% CI) Marginal R^2^/Conditional R^2^	Percentage Time in MVPA % Change (95% CI) Marginal R^2^/Conditional R^2^	Self-Reported Walking for Transport (Total Minutes) % Change (95% CI) Marginal R^2^/Conditional R^2^	Self-Reported Walking for Recreation (Total Minutes) % Change (95% CI) Marginal R^2^/Conditional R^2^	Self-Reported Overall Walking (Total Minutes) % Change (95% CI) Marginal R^2^/Conditional R^2^
Dwelling density (dwellings/Ha)	MB ^a^	**0.63 (0.37–0.89)** **0.08/0.09**	**0.47 (0.07–0.86)** **0.08/0.16**	**2.25 (0.76–3.73)** **0.09/0.16**	**3.10 (1.61–4.59)** **0.07/0.10**	**1.93 (0.60–3.25)** **0.07/0.12**
	UN ^b^	**0.82 (0.43–1.08)** **0.08/0.09**	0.65 (−0.05–1.36) 0.08/0.16	**4.58 (2.21–6.95)** **0.11/0.16**	**4.39 (2.13–6.64)** **0.07/0.10**	**4.07 (2.0–6.1)** **0.10/0.13**
	CA ^c^	**0.87 (0.24–1.49)** **0.07/0.09**	**1.20 (0.20–2.19)** **0.08/0.16**	**4.85 (1.53–8.16)** **0.09/0.17**	3.11 (−0.36–6.57) 0.06/0.09	1.81 (−1.31–4.92) 0.07/0.13
	B0500 ^d^	**1.05 (0.55–1.56)** **0.08/0.10**	**0.98 (0.15–1.80)** **0.08/0.16**	**4.88 (1.92–7.83)** **0.10/0.16**	**4.25 (1.31–7.19)** **0.06/0.09**	**3.93 (1.27–6.59)** **0.08/0.12**
	B0800 ^e^	**1.16 (0.53–1.78)** **0.07/0.09**	**1.06 (0.02–2.09)** **0.08/0.16**	**5.12 (1.38–8.86)** **0.10/0.16**	**5.59 (1.98–9.20)** **0.07/0.09**	**4.65 (1.31–7.99)** **0.08/0.12**
	B1000 ^f^	**1.17 (0.46–1.88)** **0.07/0.09**	0.83 (−0.33–1.97) 0.08/0.16	**5.20 (0.94–9.45)** **0.09/0.16**	**6.69 (2.61–10.76)** **0.07/0.09**	**4.80 (1.01–8.59)** **0.08/0.12**
	B1500 ^g^	**1.18 (0.40–1.97)** **0.07/0.09**	−0.01 (−0.76–1.82) 0.07/0.16	**6.53 (1.71–11.36)** **0.10/0.16**	**7.48 (2.86–12.10)** **0.07/0.10**	**5.75 (1.43–10.06)** **0.08/0.12**
Street connectivity (intersections/km^2^)	MB	0.02 (−0.01–0.01) 0.06/0.09	0.57 (−0.1–0.07) 0.07/0.16	0.18 (−0.14–0.51) 0.08/0.17	0.05 (−0.30–0.40) 0.05/0.10	0.10 (−0.19–0.39) 0.06/0.13
	UN	**0.27 (0.15–0.40)** **0.08/0.09**	**0.28 (0.05–0.53)** **0.09/0.16**	**1.34 (0.59–2.09)** **0.10/0.17**	**1.16 (0.39–1.93)** **0.06/0.10**	**1.04 (0.37–1.71)** **0.08/0.13**
	CA	**0.37 (0.12–0.63)** **0.07/0.09**	**0.29 (0.18–0.96)** **0.09/0.16**	**2.13 (0.75–3.52)** **0.09/0.17**	**1.50 (0.03–1.93)** **0.06/0.09**	**1.49 (0.20–2.77)** **0.07/0.13**
	B0500	**0.28 (0.09–0.47)** **0.07/0.09**	**0.27 (0.00–0.55)** **0.08/0.16**	0.94 (−0.10–1.95) 0.08/0.16	0.90 (-0.17–1.96) 0.06/0.09	**0.99 (0.07–2.37)** **0.07/0.12**
	B0800	**0.37 (0.14–0.61)** **0.07/0.09**	**0.41 (0.08–0.75)** **0.08/0.16**	0.56 (−0.73–1.85) 0.08/0.16	**2.08 (0.77–3.38)** **0.06/0.10**	**1.22 (0.07–2.37)** **0.06/0.12**
	B1000	**0.45 (0.25–0.70)** **0.07/0.09**	**0.43 (0.05–0.81)** **0.08/0.16**	**1.50 (0.05–2.94)** **0.09/0.16**	**2.07 (0.62–3.53)** **0.06/0.09**	**1.40 (0.10–2.70)** **0.07/0.12**
	B1500	**0.42 (0.19–0.65)** **0.07/0.09**	0.23 (−0.18–0.63) 0.08/0.16	**2.75 (.125–4.24)** **0.11/0.16**	**2.79 (1.35–4.22)** **0.07/0.10**	**1.53 (0.14–2.93)** **0.08/0.12**
NDAI (Score 0–31)	MB	−1.00 (−2.05–0.05) 0.06/0.09	−0.88 (−2.91–0.43) 0.07/0.16	−1.57 (−6.78–3.63) 0.08/0.17	1.18 (−4.43–6.80) 0.05/0.10	0.05 (−4.69–4.79) 0.06/0.13
	UN	−0.17 (−1.32–0.97) 0.06/0.09	1.17 (−0.81–3.14) 0.08/0.16	−3.61 (−10.06–2.83) 0.08/0.17	1.07 (−5.38–7.52) 0.05/0.10	1.15 (−4.56–6.97) 0.06/0.13
	CA	**0.88 (0.21–1.55)** **0.06/0.09**	0.41 (−0.67–1.49) 0.07/0.16	**4.64 (0.91–8.37)** **0.08/0.17**	**3.89 (0.09–7.70)** **0.06/0.10**	**4.90 (1.68–8.13)** **0.07/0.13**
	B0500	0.52 (−0.01–1.14) 0.06/0.09	0.17 (−0.64–0.97) 0.07/0.16	1.13 (−2.03–4.29) 0.08/0.17	0.92 (−2.47–4.30) 0.05/0.10	**3.36 (0.53–6.19)** **0.06/0.13**
	B0800	**0.81 (0.33–1.28)** **0.07/0.09**	**0.82 (0.15–1.49)** **0.08/0.15**	**3.51 (1.04** **–5.98)** **0.09/0.16**	1.39 (−1.29–4.07) 0.05/0.10	**4.01** **(1.82–6.20)** **0.07/0.13**
	B1000	**0.63 (0.21–1.05)** **0.06/0.09**	**0.64 (0.04–1.23)** **0.08/0.16**	**3.90 (1.76–6.05)** **0.09/0.17**	1.75 (−0.61–4.11) 0.06/0.10	**4.08 (2.18–5.97)** **0.07/0.13**
	B1500	**0.60 (0.21–0.98)** **0.07/0.09**	0.51 (−0.06–1.07) 0.08/0.16	**2.80 (0.77–4.82)** **0.09/0.17**	0.96 (−1.24–3.16) 0.05/0.10	**2.78 (0.98–4.58)** **0.06/0.13**

^a^ MB: meshblock, ^b^ UN: URBAN study neighbourhood, ^c^ CA: census area unit, ^d^ B0500: 500-m street network buffer, ^e^ B0800: 800-m street network buffer, ^f^ B1000: 1000-m street network buffer, ^g^ B1500: 1500-m street network buffer. The bold text indicates the results of models where there was evidence supporting an association between the built environment and physical activity.

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
