# Peer review of "How Do Neighbourhood Definitions Influence the Associations between Built Environment and Physical Activity?"

_ijerph, 2019, doi:10.3390/ijerph16091501_

Round 1

Reviewer 1 Report

This paper examines the relationship between built environmental qualities and physical activity patterns. The aim is to identify if certain neighbourhood definition (administrative unit or a network buffer) or measure on built environment characteristics is more relevant in terms of being physically active (also measured with several variables). The results, as the authors predict, show that the relationship between physical environment characteristics and physical activity patterns depend on the definition of neighbourhood and the built characteristics assessed, with most consistent results using a 800-meter road network buffer.

The paper is overall nicely written and mostly easy to follow. The introduction is clear and grounds the study well. However, my main concern is that I cannot evaluate the statistical analyses because there is a lot to improve in terms of how they are described and reported (and interpreted, to some extent). I will explain my concerns in the following paragraphs.

First, Section 2.6 on statistical analyses: it is good that the authors tested whether the mixed effects model was appropriate but it is unclear how they did this. On L240 they say that this was done by comparing an empty model with and without clustering taken into account, but in the next sentence they state that “model’s fit was significantly improved with the inclusion of the neighbourhood level variables, so the multilevel structure was maintained”. Which of these was the actual criteria for choosing the use the multilevel model? And does the neighbourhood-level variables refer to all three built environment measures, or the adjusted factors as well? Intra-class correlations are a good and easier-to-grasp measure that helps to understand the effect of the clustering, and I strongly recommended that the authors add these in the descriptive tables.

Second, the role of the adjusted individual and neighbourhood factors in the analysis is unclear. Based on Section 2.6 and Table 5 one cannot evaluate what the ‘fully-adjusted’ in the table caption refers to – do all the models have the same explanatory variables or not? If not, the models are not comparable, and the reader does not even know which factors are adjusted for in each model. I strongly recommend the authors to report and discuss models that have the exact same set of explanatory variables, regardless of their significance levels. It is far more important that they are theoretically justified and that the more important factors in the models, namely the built environment measures and the neighbourhood definitions, are comparable across models. There is also no need to explain how the adjustments were conducted in steps (L251-255) if the results from these different steps are not reported. Furthermore, the coefficients for the adjusted factors may be interesting for the reader and potentially relevant for meta-analytic purposes. I understand reporting them in each model might just add confusion due to large amount of analyses but it is a good practice to report these as appendices. Also, for the sake of this review, I would have preferred to examine them to understand what was actually modeled (e.g. at the current version there is no way to properly assess the validity of the statement at lines 291-292).

Thirdly, Table 5 should be improved. In addition to the fact that it was missing from the version I originally received, the table should be more reader-friendly. What are the different R2 values – fixed/fixed+random, as indicated in the analysis section? This should be clearly stated. It would also help the reader to shortly remind what is included in the random effect – is it only the built-environment measure (in the first column) or also the other neighbourhood-level predictors, socioeconomic deprivation and preference? Why are some cells highlighted and in bold? I am assuming this indicates a difference from 0 but it should nevertheless be clearly stated in table caption; these should be self-explanatory. It is not necessary to both bold and highlight cells if their information value is exactly the same; bolding is a more traditional way to indicate a difference from 0.

Fourth and finally regarding the results, the authors constantly state that the most consistent pattern of a positive association is found with the 800m buffer, but looking at Table 5 the 1000m buffer shows equally many positive associations, with mostly similar estimates and CIs. Based on this table, I see no point in highlighting network buffers of 1000m as equally relevant in terms of physical activity throughout the paper.

In addition I have some minor issues and suggestions to make:

L226: why was this measure chosen, any scientific grounds for this?

L247: why were the outcomes log-transformed?

Table 2: What is the time frame for these variables – mean per hour/day/week?

L435: maybe a short definition or a reference to ‘kernel density measures’ would be useful for a reader that is unfamiliar with this concept

Table 5: cell highlights definition missing in the caption

Overall, I see a great value in this paper but in its current form I find it hard to evaluate its scientific appropriateness in terms of the methods. These issues might be resolved with clearer reporting.

Author Response

Response to Reviewer 1 

Thank you for your thoughtful comments that have helped us improve the communication of our methods and results. 

First, Section 2.6 on statistical analyses: it is good that the authors tested whether the mixed effects model was appropriate but it is unclear how they did this. On L240 they say that this was done by comparing an empty model with and without clustering taken into account, but in the next sentence they state that “model’s fit was significantly improved with the inclusion of the neighbourhood level variables, so the multilevel structure was maintained”. Which of these was the actual criteria for choosing the use the multilevel model? And does the neighbourhood-level variables refer to all three built environment measures, or the adjusted factors as well?  

Thank you. We used the likelihood ratio test to compare empty models with and without the clustering variables (city, and urban neighbourhood). Based on the results of the likelihood ratio test (p<0.001) we decided to include the clustering variables and maintain a multilevel model. We have clarified the text as shown below. New text is shown in bold. 

The associations between the built environment and physical activity measures were modelled using linear multi-level mixed effect models to take into account the clustering of individuals within neighbourhoods (defined as the URBAN study neighbourhood) and cities. The multi-level fixed effect model was chosen to assess the effect of neighbourhood characteristics on individual physical activity level. The appropriateness of the multilevel structure was tested by applying. T the likelihood ratio (LR) test was used to compare an empty model with and without adjustment for clustering (URBAN study neighbourhoods nested in cities). The model’s fit was significantly improved with the inclusion of the neighbourhood level variables (p < 0.001), so the multilevel structure was maintained. The association between built environment exposures and physical activity were assessed through three models and progressively adjusted for confounders. 

Intra-class correlations are a good and easier-to-grasp measure that helps to understand the effect of the clustering, and I strongly recommended that the authors add these in the descriptive tables. 

Thank you for this thoughtful suggestion, as suggested we have also reported the ICC values for the null models in Table 2 below: 

Table 2. Descriptive statistics for the physical activity outcome measures assessed over a 7 day period. 

Low walkability 

High walkability 

Physical activity outcome 

Mean 

Median 

SD 

Mean 

Median 

SD 

Adjusted ICC for null model 

Self-reported walking for transport (total minutes) 

80.0 

40 

125.2 

109.4 

50 

154.7 

0.136 

Self-reported walking for recreation (total minutes) 

82.5 

30 

125.0 

83.8 

30 

128.6 

0.011 

Self-reported overall walking (total minutes) 

161.8 

100 

191.6 

192.4 

120 

220.5 

0.120 

Mean accelerometer counts per hour 

8701.1 

8040.0 

4215.0 

9426.6 

8586.7 

4692.6 

0.048 

% time spend in MVPA 

12.3 

11 

6.6 

12.5 

11 

6.9 

0.080 

Second, the role of the adjusted individual and neighbourhood factors in the analysis is unclear. Based on Section 2.6 and Table 5 one cannot evaluate what the ‘fully-adjusted’ in the table caption refers to – do all the models have the same explanatory variables or not? If not, the models are not comparable, and the reader does not even know which factors are adjusted for in each model. I strongly recommend the authors to report and discuss models that have the exact same set of explanatory variables, regardless of their significance levels. It is far more important that they are theoretically justified and that the more important factors in the models, namely the built environment measures and the neighbourhood definitions, are comparable across models.  

Thank you for identifying issues in our reporting of the key results of the paper. All models reported in the paper have the same set of covariates/explanatory variables. For each built environment measure tested, the reported models varied only by neighbourhood definition. We have clarified this in the methods and results table caption as shown below. 

The association between each of the three built environment measures and each of the five physical activity measures was modelled separately for each of the seven neighbourhood definitions. Each association was assessed by adjusting for individual-level factors (sex, age, ethnicity, income, marital status, education, employment, and car access), neighbourhood socioeconomic deprivation, and neighbourhood preference. All models reported in this paper are fully adjusted for these covariates. 

Table 5. Percentage change (95% CI) in fully adjusted models of physical activity, for a 1 unit change in the built environment measures for the seven neighbourhood boundaries. All models are fully adjusted for sex, age, ethnicity, income, marital status, education, employment, car access, neighbourhood socioeconomic deprivation, and neighbourhood preference. Bold text indicates results where there was some evidence (confidence intervals did not cross zero) to support an association between the built environment and physical activity. 

There is also no need to explain how the adjustments were conducted in steps (L251-255) if the results from these different steps are not reported.  

We have deleted this text as suggested. 

Adjustment variables were added in steps. The first versions of the models included sex, age, ethnicity, income, marital status, education, employment, and car access; these a priori variables were retained in the model regardless of their significance. Next neighbourhood socioeconomic deprivation variable was added, and finally, the neighbourhood preference variable was added.   

Furthermore, the coefficients for the adjusted factors may be interesting for the reader and potentially relevant for meta-analytic purposes. I understand reporting them in each model might just add confusion due to large amount of analyses but it is a good practice to report these as appendices. Also, for the sake of this review, I would have preferred to examine them to understand what was actually modeled (e.g. at the current version there is no way to properly assess the validity of the statement at lines 291-292). 

This is a very good point. We did not include the coefficients due to the challenges in presenting results from many models. Therefore, we have added supplementary tables that contain this information. Please see the attached supplementary document. 

Thirdly, Table 5 should be improved. In addition to the fact that it was missing from the version I originally received, the table should be more reader-friendly. What are the different R2 values – fixed/fixed+random, as indicated in the analysis section? This should be clearly stated. It would also help the reader to shortly remind what is included in the random effect – is it only the built-environment measure (in the first column) or also the other neighbourhood-level predictors, socioeconomic deprivation and preference? Why are some cells highlighted and in bold? I am assuming this indicates a difference from 0 but it should nevertheless be clearly stated in table caption; these should be self-explanatory. It is not necessary to both bold and highlight cells if their information value is exactly the same; bolding is a more traditional way to indicate a difference from 0. 

Thank you for your suggestions to improve the readability of Table 5. We have made the following changes: 

We have clarified which variables were modelled as fixed and random effects. Edited results text as shown below. 

The results of the fully adjusted models presented in Table 5 indicate that whether or not an association between the built environment and physical activity was detected depends on the choice of neighbourhood definition, built environment measure, and physical activity measure. For all models, the URBAN study neighbourhood and city were modelled as random effects and all other explanatory variables were modelled as fixed effects. The results are reported as the percentage change in the physical activity measure unit per unit increase in the built environment measure (Table 5). Bold text indicates results where there was some evidence (confidence intervals did not cross zero) to support an association between the built environment and physical activity. As expected in built environment research, the effect sizes were small as individual outcomes are more strongly associated with individual predictors. 

We have clarified the R2s by editing the first row of Table 5 as shown below (examples for two columns only). 

Mean accelerometer counts/hour 

% change (95% CI) 

marginal R2/conditional R2 

Percentage time in MVPA 

% change (95% CI) 

marginal R2/conditional R2 

We have removed the highlighting. 

We have added text to the table caption to describe the meaning of the bold text 

Table 5. Percentage change (95% CI) in fully adjusted models of physical activity, for a 1 unit change in the built environment measures for the seven neighbourhood boundaries. All models are fully adjusted for sex, age, ethnicity, income, marital status, education, employment, car access, neighbourhood socioeconomic deprivation, and neighbourhood preference. Bold text indicates results where there was some evidence (confidence intervals did not cross zero) to support an association between the built environment and physical activity. 

Fourth and finally regarding the results, the authors constantly state that the most consistent pattern of a positive association is found with the 800m buffer, but looking at Table 5 the 1000m buffer shows equally many positive associations, with mostly similar estimates and CIs. Based on this table, I see no point in highlighting network buffers of 1000m as equally relevant in terms of physical activity throughout the paper. 

Thank you. We agree and have adjusted the text throughout as follows. 

Abstract 

Results demonstrated that, while there was no single ideal neighbourhood definition, the built environment was most consistently associated with a range of physical activity measures when the 800 m and 1000 m road network buffers was were used. For the street connectivity and destination accessibility measures, associations with physical activity were less likely to be detected at smaller scales (less than 800 m). In line with some previous research, this study demonstrated that the choice of neighbourhood definition can influence whether or not an association between the built environment and adult’s physical activity is detected or not. This study additionally highlighted the importance of the choice of built environment attribute and physical activity measures.  While we identified the 800 m and 1000 m road network buffers as the neighbourhood definitions…. 

Results 

Overall, associations between the built environment and physical activity measures were most consistently detected when the 800 m and 1000 m road network buffers was were used. 

Discussion 

However, our study showed that associations between the built environment and physical activity were most consistently detected when the 800 m and 1000 m road network buffers were was used.   

Conclusion 

The 800 m and 1000 m road network buffers werewas the neighbourhood definitions where associations between the built environment and physical activity were most consistently detected. 

5) L226: why was this measure chosen, any scientific grounds for this? 

We measured neighbourhood preference so that we could adjust for this potential confounder. We have edited this section as shown below. 

Individuals may choose to live in neighbourhoods that support physical activity, introducing the possibility that individual neighbourhood preference may confound associations between neighbourhood environments and physical activity [27]. Therefore, Nneighbourhood preference was measured using items developed by Levine et al. [37]. 

L247: why were the outcomes log-transformed? 

We have edited the text as follows: 

All outcome variables were log transformed to have a normal distribution and aid comparison across models. 

7) Table 2: What is the time frame for these variables – mean per hour/day/week? 

We have clarified this in the table caption: 

Table 2. Descriptive statistics for the physical activity outcome measures assessed over a 7 day period. 

8) L435: maybe a short definition or a reference to ‘kernel density measures’ would be useful for a reader that is unfamiliar with this concept 

We have added additional text to elaborate on kernel density measures: 

For example, kernel density measures are an underutilized technique in built environment and health research that account for the proximity of built environment features to one another [69]. , yYet in a recent food environment study they showed stronger associations 

9) Table 5: cell highlights definition missing in the caption 

We agree that it is not necessary to have bold and highlights, so have removed cell highlights. 

Reviewer 2 Report

In general, this is an interesting paper which helps us understand how the neighborhood definitions of different types and geographical scales influence the association between the built environment and physical activity. The manuscript was well written and limitation also was sufficiently acknowledged. 

1. Table 5 is not presented in the present manuscript. I cannot make any comments about this part of the results.

2. Could the descriptive statistics for physical activity and built environment attributes (from table 2 to table 4) be presented stratified by walkability of the neighborhood (24 low vs. 24 high) or by suburban, neighborhood activity Centre, and CBD neighborhood?

Author Response

Response to Reviewer 2 

Thank you for your comments and time spent reviewing our manuscript. 

Table 5 is not presented in the present manuscript. I cannot make any comments about this part of the results 

We apologise that Table 5 was not included in your version. It has been added and we include the revised table at the end of this document for your reference. 

2. Could the descriptive statistics for physical activity and built environment attributes (from table 2 to table 4) be presented stratified by walkability of the neighborhood (24 low vs. 24 high) or by suburban, neighborhood activity Centre, and CBD neighborhood? 

We have added stratification by walkability to Table 2, and also ensured that these descriptive statistics are reported for a consistent time period (over 7 days). The new table is shown below. 

Table 2. Descriptive statistics for the physical activity outcome measures assessed over a 7 day period. 

Low walkability 

High walkability 

Physical activity outcome 

Mean 

Median 

SD 

Mean 

Median 

SD 

Adjusted ICC for null model 

Self-reported walking for transport (total minutes) 

80.0 

40 

125.2 

109.4 

50 

154.7 

0.136 

Self-reported walking for recreation (total minutes) 

82.5 

30 

125.0 

83.8 

30 

128.6 

0.011 

Self-reported overall walking (total minutes) 

161.8 

100 

191.6 

192.4 

120 

220.5 

0.120 

Mean accelerometer counts per hour 

8701.1 

8040.0 

4215.0 

9426.6 

8586.7 

4692.6 

0.048 

% time spend in MVPA 

12.3 

11 

6.6 

12.5 

11 

6.9 

0.080